# International Consensus Guidelines on the Safe and Evidence-Based Practice of Mesotherapy: A Multidisciplinary Statement

**DOI:** 10.3390/jcm14134689

**Published:** 2025-07-02

**Authors:** Massimo Mammucari, Domenico Russo, Enrica Maggiori, Marco Rossi, Marzia Lugli, Raffaele Di Marzo, Alberto Migliore, Raimondo Leone, Kamil Koszela, Giustino Varrassi

**Affiliations:** 1Italian Society of Mesotherapy, ASL RM 1, 00165 Rome, Italy; dt.domenico.russo@live.it (D.R.); enricamaggiori@libero.it (E.M.); dimarzoraffaele@gmail.com (R.D.M.); 2Department of Anesthesia, Intensive Care and Pain Medicine, Catholic University Holy Heart, Policlinico A. Gemelli, 00168 Rome, Italy; marco.rossi1@unicatt.it; 3Department of Cardiovascular Surgery, Hesperia Hospital, 41125 Modena, Italy; lugli@chirurgiavascolaremodena.com; 4San Pietro Fatebenefratelli Hospital, 00189 Rome, Italy; migliore.alberto60@gmail.com; 5Department of Public Health and Life Science, Catholic University of the Sacred Heart, 00168 Rome, Italy; raimondo.leone@unicatt.it; 6Department of Neuroorthopedics and Neurology Clinic and Polyclinic, National Institute of Geriatrics, Rheumatology and Rehabilitation, 02-637 Warsaw, Poland; kamil.aikido@interia.pl; 7Fondazione Paolo Procacci, 00193 Rome, Italy; giuvarr@gmail.com

**Keywords:** mesotherapy, guideline, pain, phlebology, dermo-esthetic, sport medicine

## Abstract

**Introduction.** Mesotherapy is a widely used technique around the world. However, there is currently no internationally recognized, evidence-based standard for its various clinical applications. To address this gap, we have reviewed the current state of the art, critically evaluated its clinical benefits and limitations, and proposed a set of standards including procedural steps, recommended actions, and practical instructions in the form of protocols, guidelines, and expert recommendations. **Methods.** A team of researchers conducted a comprehensive literature review, selecting studies published between 1976 and 2023. Drawing on the available evidence and the needs expressed by patient associations, 23 clinical questions were developed and presented to a panel of experts. Through multiple rounds of evaluation, evidence-based recommendations were formulated and subsequently submitted for structured evaluation and voting by a broad, multidisciplinary panel of international experts, representing numerous national and international scientific societies. **Results.** The recommendations outlined in this guideline support the use of mesotherapy across diverse clinical and organizational settings, providing a standardized framework that ensures both efficacy and patient safety in osteoarticular pain, rehabilitation, and dermatological fields. **Conclusions.** The era of mesotherapy based on personal beliefs now gives way to evidence-based practice. The limitations underscore the need for continued high-quality research and scheduled guideline updates.

## 1. Introduction

In 2025, the Italian Society of Mesotherapy (SIM) celebrates its 50th anniversary. SIM has continuously aimed to understand the advantages and limitations of mesotherapy through critically analyzing the available evidence. Today, mesotherapy represents a therapeutic option capable of achieving clinical effects with low-dose active substances [1].

Despite numerous publications confirming its efficacy and safety in various clinical conditions, a standardized protocol for its execution has yet to be established [2]. The need for defining a standard may seem restrictive, as it excludes certain applications that do not meet the scientific threshold. However, it is precisely these applications, lacking supporting data, that stand to benefit most from future research. Developing a standard is therefore crucial to mesotherapy being applied as an interesting and futuristic therapeutic tool in patient care. Accordingly, SIM had previously formulated an initial set of evidence-based recommendations to support this objective [3]. Following suggestions from national health authorities, which urge scientific societies to regularly update their guidelines [4], a new and broader systematic review process of the available evidence has been conducted. Although mesotherapy has been used successfully for decades worldwide, there is still a need for standardization in its clinical practice to minimize the risks associated with any infiltrative therapy, underlining the importance of a universally shared guideline. This review is thus pivotal in guiding both future research and current clinical practices, with the aim being to establish an international standard to integrate this technique into treatment and care pathways.

## 2. What Is New in This Recommendation

The key strength of this document, compared to previous recommendations, lies in its broader representativeness. For its development, international experts from 16 countries, representatives of mesotherapy scientific societies, were involved. In addition, specialists from various disciplines and representatives of 15 national scientific societies participated, contributing to a more multidisciplinary perspective. From the very beginning of the drafting process, representatives from four patient associations were involved. Their participation helped align the guideline with patients’ real needs and ensure it considered different care settings [5]. To eliminate biases stemming from personal beliefs and to facilitate future updates, a standardized procedure was followed in developing this recommendation [6]. The scientific process also considered unanswered questions due to a lack of data, identified debated topics, and pinpointed areas with limited scientific evidence, all aimed at shaping future research directions. The contents of this document therefore represent a starting point for applying mesotherapy and for planning further scientific research.

## 3. Methods

The Bibliographic Research Committee reviewed all Medline literature using the keyword ‘mesotherapy,’ based on which the Steering Committee selected the most relevant studies and, in consultation with patient and volunteer associations, identified 23 key clinical questions (Table 1). These questions were formulated through a process involving a literature review, patient needs, and the potential applications of mesotherapy in the various areas explored to date. Each question was submitted to a group of experts (Editorial Committee). After several rounds, the responses and related recommendations were collected. The recommendations that reached final approval were then submitted for voting by both national and international experts. The voting process included representatives from mesotherapy scientific societies in other countries (International Scientific Committee) as well as from several national scientific societies (Representatives of National Scientific Associations). The level of evidence for each recommendation was assessed using the methodology proposed by the Agency for Health Care Policy and Research (AHCPR), based on the types of studies available (Table 2). The recommendations were voted on using a 5-point Likert scale: strongly disagree (1), disagree (2), neutral (3), agree (4), strongly agree (5). The pre-established consensus threshold for a recommendation to be considered approved was set at a minimum of 70% agreement, as determined by the commission to enhance the safety and effectiveness of the methodology. After processing the expert voting results, a final document was drafted and reviewed by four external reviewers (Figure 1). It was then reviewed by four external reviewers using the AGREE quality of reporting checklist [7]. This article presents the final version of the document, as approved by the reviewers.

### 3.1. Question 1: How Should Mesotherapy Be Defined?

The first intracutaneous injections described for analgesic purposes date back to the mid-1800s [8].

The term “mesotherapy” was firstly used in 1958 by Michel Pistor, who later described the method as a subcutaneous therapy useful in various clinical situations [9]. Mesotherapy spread worldwide for the treatment of numerous conditions, but it was practiced in different ways and, in some cases, considered a form of alternative medicine [10]. In 2004, Sergio Maggiori proposed the term “Local Intradermal Therapy” (LIT) to emphasize the role of injecting active ingredients into the superficial layer of the skin [11,12]. However, due to the use of injected products without a scientific rationale, the technique faced criticism, particularly when applied in dermatology [13,14]. Consequently, in 2011, the Italian Society of Mesotherapy initiated an important review process. In a first position paper, the following definition was proposed: “Minimally invasive technique based on microinjections of active ingredients into the surface layer of the skin corresponding to the area to be treated” [1]. In 2014, an international consensus revised this definition as follows: “A technique based on the administration of pharmaceutically active substances in the upper layers of the skin” [15]. In 2021, a new international consensus introduced two new definitions: “The term mesotherapy means the infiltration into the superficial layer of the skin for preventive, curative, or rehabilitative purposes” and “The term local intradermal therapy refers to a series of dermal micro deposits of products that result in their slow diffusion into the underlying tissues.” [3]. This latest position endorses the infiltration of pharmaceutically active substances and includes the now widespread use of medical devices, particularly in dermatology. Nonetheless, the term “mesotherapy” has been retained alongside the more specific term “local intradermal therapy.” In fact, mesotherapy has always been regarded as a localized treatment, and many studies have compared it with systemic therapies, whether oral or parenteral. In contrast, the subcutaneous administration of active substances should be considered a systemic parenteral treatment, commonly used in various clinical areas such as palliative care, oncology, and immunotherapy [16]. Indeed, it has already been noted that during mesotherapy, a portion of the injected drug is absorbed into the bloodstream, producing a systemic effect that adds to the local effect [3]. Future studies will be tasked with determining the roles played by the local and systemic effects resulting from mesotherapy administration [17]. In conclusion, having evaluated previous definitions, the experts strongly recommend the definitions reported in Table 3 (each such table reports the recommendation; the questions (Q) and the related answers from which the recommendations were obtained; the level of the studies considered (according to AHCPR); the mean ± SD of the score obtained from the experts’ vote; and the percentages of agreement, abstentions, and disagreement).

### 3.2. Question 2: How Is Mesotherapy Performed?

For years, the mesotherapy technique was passed down from one physician to another. Only in 2011 did a consensus of experts propose standardized recommendations for performing mesotherapy [1], suggesting the systematic use of a 4 mm long needle with a 0.4 mm gauge (27 G), or a 13 mm long needle (30 G or 32 G), with angle adjustments depending on the skin area. Each injection should contain no more than 0.1–0.2 mL of product, and microinjections should be spaced 1 to 3 cm apart. Mesotherapy involves a series of microinjections in a specific area of the skin. Injection depth has been a topic of much debate, but the highest level of consensus was reached by defining the standard injection depth as within 2 mm (Figure 2) [12,18]. A 30° angle relative to the skin surface facilitates an injection depth of approximately 2 mm. By increasing or decreasing the angle, different injection depths can be achieved. These recommendations were also intended with the objective to standardize the technique with the objective to facilitate comparisons of clinical data across studies. As a result, the use of multi-injectors is discouraged because they do not allow the physician to achieve a consistent injection depth with each needle or the precise targeting of skin points previously selected by the physician (e.g., trigger points). Furthermore, with multi-injectors it is not possible to avoid nevi or other skin lesions that should not be infiltrated. Moreover, multi-injectors do not allow microinjections to be spaced at variable distances [1]. The recommendation to inject different substances using different syringes and at different inoculation sites [3] has been strongly reaffirmed and validated during this approval process. Adherence to the aseptic technique is a fundamental recommendation to avoid the adverse events reported in the literature [19], often incorrectly attributed to mesotherapy but actually caused by a failure to follow aseptic guidelines (Table 4).

### 3.3. Question 3: What Is the Mechanism of Action of Mesotherapy?

Preclinical studies have shown that the intradermal injection of NSAIDs, anesthetics, and antibiotics results in higher and more prolonged concentrations in the tissues beneath the injection site (skin, muscles, and joints), compared to intramuscular administration. This supports the hypothesis of a persistent local pharmacological action [1,20,21,22].

It has also been reported that the analgesic effect of mesotherapy is achieved using lower doses of medication administered at longer intervals compared to systemic (oral and parenteral) routes [23,24,25,26,27,28,29]. This includes the effect of prolonged systemic absorption due to the slow uptake from intradermal drug depots [1,8]. The most commonly used drugs for managing localized pain include anesthetics, muscle relaxants, analgesics, and anti-inflammatory agents. It is therefore possible to hypothesize that interactions with the immune system, the endorphin system [30,31,32], and the network of glial cells abundantly present in the dermis [33,34] and the participation of keratinocytes in nociceptive activity all support the hypothesis of dermal modulation triggered by mesotherapy. Some evidence suggests that the mechanical stimulation caused by multiple needle pricks may also play a role [35]. In fact, when comparing two groups of patients treated with pharmacological mesotherapy or dry needling, both treatments resulted in pain reduction [18,36]. Injection points may also influence outcomes, as some data suggest that mesotherapy applied to trigger points is more effective in reducing localized pain [37]. The local analgesic effect may partly result from tissue stretching and the chemical–physical changes induced by the injected liquid in the dermis. Mesotherapy performed with saline solution reduces pain, albeit to a lesser degree and for a shorter duration compared to mesotherapy with drugs [38,39]. Moreover, saline solution has proven to be less effective than sterile water, suggesting a different intradermal chemical–physical effect [40,41]. In a randomized study, sterile water was found to be more effective than dry needling [42], suggesting that the analgesic effect may be induced by osmotic draw and increased pressure in the tissues, which possibly activates afferent nerve fibers (A-delta and C fibers) and the gate control mechanism [43]. It should also be noted that the delayed systemic action of the injected active ingredient may prolong the effects over time and explain the reduced frequency of mesotherapy sessions (generally weekly). Finally, within such a complex system of hypotheses, a component of placebo effect must also be considered, as in any other analgesic therapy [44,45,46]. In conclusion, the local pharmacological effect, microinjections induced by the needle, the chemical–physical stimulation from the injected liquid volume, stimulation of specific trigger points, and additional neuroimmune-mediated actions all appear to contribute to the clinical response observed after mesotherapy administration. The collection of these hypothesized effects has been termed “mesodermal modulation” (Figure 3, Table 5).

### 3.4. Question 4: What Is the Advantage of the Drug-Sparing Effect of the Intradermal Route in Immunoprophylaxis?

The drug-sparing effect observed in analgesic applications through intradermal administration has also been reported in the field of immunology. Numerous studies confirm that higher antibody titers can be achieved with lower antigen doses via the intradermal route, compared to deeper administration routes [47,48,49,50,51,52,53,54]. A recent narrative review highlights the pharmacoeconomic benefits of intradermal vaccination for various pathogens, including influenza virus, rabies virus, poliovirus (PV), hepatitis B virus (HBV), hepatitis A virus (HAV), diphtheria–tetanus–pertussis (DTP) bacteria, human papillomavirus (HPV), Japanese encephalitis virus (JE), meningococcus, varicella-zoster virus (VZV), and yellow fever virus [55] (Table 6).

### 3.5. Question 5: Which Substances Are Injected?

The choice of drugs depends on the diagnosis, and the selected active ingredients must be those approved for managing the specific condition or symptoms being treated. Drugs not authorized for the condition in question should be used only in an experimental setting, and off-label use should be limited to substances for which high-quality clinical studies already exist that demonstrate their tolerability and efficacy (3). Some drugs may not be approved for intradermal or subcutaneous use, and in such cases, the physician must follow the appropriate off-label use protocols. Several studies have been conducted using drug mixtures, showing good tolerability and efficacy. However, due to the heterogeneity of the mixtures and the lack of comparisons between individual components, it is not possible to recommend mixtures as more effective than single-drug administration. Only one randomized controlled trial (RCT) has compared the analgesic effect of mesotherapy using a single drug versus a combination of two drugs in patients with spinal pain. Both the mixture and the single-drug approach resulted in pain reduction, with no significant differences [25] (Table 7). Further studies comparing the efficacy and safety of individual drugs and/or mixtures of different drugs are recommended [17].

### 3.6. Question 6: Can Mesotherapy Be Included in the Management of Patients with Localized Musculoskeletal Pain?

The intradermal route was initially tested for analgesic purposes, and mesotherapy became widely used for this indication. However, the first review on the use of mesotherapy in treating musculoskeletal pain dates back to 2012 [23]. Some RCTs showed that mesotherapy was superior to a placebo [56] and to laser therapy combined with NSAIDs [57], and non-inferior to systemic ketoprofen plus methylprednisolone [58] for low back pain. Mesotherapy with lidocaine injected into acupuncture points proved to be more effective than lidocaine at trigger points [37], and it was also effective in treating calcific tendinitis of the shoulder using EDTA and procaine [59]. The combination of TENS and mesotherapy improved the analgesic response in cervicobrachial pain [60]. A more recent review analyzed seven studies: two on knee osteoarthritis and five on spinal pain [26]. The authors concluded that mesotherapy is well tolerated and effective in reducing both acute and chronic musculoskeletal pain. However, the treatments involved different drugs, and it was not possible to determine which was the most effective. An additional meta-analysis confirmed that mesotherapy is safe and more effective than systemic therapy for localized pain and functional limitations due to various musculoskeletal conditions [61]. Nonetheless, heterogeneity was observed in the substances used, injection techniques, associated treatments, session frequency, and number of sessions. Regarding the speed of clinical response after mesotherapy, two RCTs highlighted a relatively rapid onset compared to systemic routes. In fact, emergency room patients with low back pain treated with lidocaine, thiocolchicoside, and tenoxicam via mesotherapy showed greater improvement than those who received intravenous dexketoprofen after 15, 30, and 60 minutes and 24 hours [24]. Similarly, patients with acute trauma treated in the emergency department with tenoxicam-based mesotherapy experienced greater pain relief at 10, 30, 60, and 120 minutes compared to those who received intravenous dexketoprofen [27]. A review based on previously published studies defined mesotherapy as a viable option for managing musculoskeletal pain [3]. Two subsequent RCTs investigated its effect on knee pain. One study compared systemic administration to mesotherapy, showing similar pain reduction on the VAS at 2, 4, and 8 weeks, but mesotherapy scored higher on the Oxford Knee Score (OKS) and the Persian version of the WOMAC index [62]. A second study compared patients with gonarthrosis treated with mesotherapy using a drug mixture to a group treated with saline solution. Both groups improved, but the drug mixture group exhibited significantly better outcomes at 8 and 16 weeks [63]. An analysis of the available data shows that authors tend to manage localized pain with mesotherapy using primarily NSAIDs, anesthetics, and muscle relaxants, which appear to offer clinical benefit at lower doses than those required for systemic administration. In RCTs, NSAIDs and anesthetics have been used alone or in combination, but combinations have not been directly compared to the individual components. No conclusion can therefore be drawn in favor of combinations over single drugs. For the same reason, conclusions cannot be made regarding the various mixtures used. Some features of mesotherapy that have particularly appealed to patients include the lower drug doses required to achieve therapeutic goals and the reduced frequency of administration (typically weekly). This is supported by findings in several studies (Table 8).

It should be noted that the intradermal injection of saline solution also leads to pain reduction, albeit to a lesser extent than with drugs [38,39,70], and that sterile water is more effective than isotonic solution [64,71] or needle puncture alone [42]. This suggests that mesotherapy performed with saline solution or sterile water should not be considered a placebo when compared to other treatments. The use of mesotherapy during childbirth deserves specific attention. A systematic review on the use of intradermal or subcutaneous injections for contraction pain and back pain during labor has explored the topic [72]. Seven studies (involving 766 patients) were considered: four used the intradermal route, two the subcutaneous route, and one used both routes. The authors concluded that there is limited conclusive evidence supporting the effectiveness of injectable sterile water for labor pain. However, in a randomized controlled trial involving 168 patients, intradermal injections of sterile water were found to be superior to needle puncture alone in managing severe labor-related back pain at 30, 60, 120, and 180 minutes after administration. Randomized controlled clinical trials are warranted to evaluate not only the clinical effectiveness but also the economic benefits, as lower doses and reduced frequency of administration are factors that contribute to greater patient satisfaction (Table 9).

### 3.7. Question 7: Can Mesotherapy Be Integrated into the Individual Rehabilitation Plan (IRP)?

The role of mesotherapy in rehabilitation is closely linked to its ability to reduce pain, whether acute, subacute, or chronic. Given the numerous studies in this area, it is entirely plausible that better pain control facilitates the earlier and more effective implementation of rehabilitation therapy [23]. A review of seven studies (see blue section of Table 8) concluded that “mesotherapy showed a good effect to reduce acute and chronic musculoskeletal pain and, also, it is a well-tolerated treatment” [26]. A 2021 consensus reported that “mesotherapy can be considered in the individual path of rehabilitation” [3]. Following this review, an observational case–control study of 78 patients assessed the efficacy of mesotherapy combined with rehabilitation exercise in patients with fibromyalgia. All patients underwent a rehabilitation program including exercise + TENS + CO_2_ laser and received mesotherapy either with diclofenac + thiocolchicoside + mepivacaine (treatment group) or saline solution (placebo group). Both groups experienced significant pain reduction, as measured by the Visual Analog Scale (VAS), although the improvement was more marked in the group treated with active drugs. A significant reduction in the Neck Disability Index (NDI) and improvement in quality of life (measured with SF-12) was also observed in the group treated with mesotherapy using active drugs. The authors state that the treatment “is a safe and effective procedure in the management of neck pain in fibromyalgia patients in the short term, regarding pain reduction, functional recovery, and quality of life” [69]. Mesotherapy has also been proposed as part of an integrated rehabilitation approach for Web Axillary Pain Syndrome [73], and in comparison with TENS, it showed a reduction in analgesic consumption [74] (Table 10).

### 3.8. Question 8: Can Sports Injuries Benefit from Mesotherapy?

Localized pain, muscle contracture, and reduced joint function are frequent symptoms in sports-related injuries. For this reason, mesotherapy is applied in the fields of exercise and sports medicine as well as in the rehabilitation of athletes. Among the most reported benefits are shortened functional recovery times and pain reduction, achieved with a relatively small number of weekly sessions, which help facilitate rehabilitation and improve athlete compliance (Table 11).

Over the years, various pharmacological strategies have been used; however, it is important to emphasize that, although mesotherapy relies on the local action of drugs injected through a series of microinjections, systemic absorption, albeit delayed, must also be considered. Some of the substances used in mesotherapy practice are included in the WADA (World Anti-Doping Agency) list, particularly corticosteroids. WADA provides details on minimum washout periods, which for local injections range from 3 to 10 days [83]. Since no specific studies are available to establish the washout period for corticosteroids administered via mesotherapy, it is necessary to follow the same washout periods as that for intramuscular administration. Considering the evidence supporting pain management and rehabilitation applications, the use of mesotherapy in managing athletes after sports injuries is plausible, although strong evidence and high-quality studies are still lacking (Table 11).

### 3.9. Question 9: Can Mesotherapy Be Included in the Care Pathway for the Signs and Symptoms of Chronic Venous Disease (CVD) and the Prevention of Its Complications (PEFS)?

The term chronic venous disease (CVD) refers to “morphological and functional abnormalities of the venous system of long duration, manifesting with symptoms and/or signs that require investigation and/or care” [84]. It is classified using the CEAP system (Clinical–Etiology–Anatomy–Pathophysiology) [85], and it refers to pathological hemodynamic alterations in the lower limbs that cause a wide range of symptoms and signs (telangiectasias, reticular veins, varicose veins, skin changes, and chronic venous leg ulcers). Vascular changes in CVD also contribute to the pathophysiology of edematous-fibrosclerotic panniculopathy (PEFS) through the deposition of altered glycosaminoglycans on the capillary walls, resulting in increased capillary–venular permeability and fluid leakage into the dermis. Persistent edema, vascular congestion, and tissue hypoxia promote the thickening and increase in number of interlobular septa in the dermis and adipose tissue, leading to the appearance of depressed skin areas due to disrupted dermal architecture, hallmark signs of PEFS [86]. A linear correlation has been described between venous stasis symptoms and signs in chronic venous disease (CVD) of the lower limbs and those of PEFS (heaviness, pain, cramps, edema, telangiectasias) [87]. Recent guidelines recommend conservative management in symptomatic CVD (C0–C5), and while awaiting eventual surgical intervention [84]. An expert consensus recommends initiating treatment even before clinical signs appear, to slow the progression and improve the quality of life [88]. The rationale for using mesotherapy in CVD and PEFS was clinically tested as early as the 1980s (data on file of SIM regarding 1,041 patients studied in open-label or retrospective studies). More recently, several clinical studies have further explored the role of mesotherapy in treating venous insufficiency of the lower limbs, using active ingredients with antithrombotic, profibrinolytic, and anti-inflammatory properties [89,90]. Additional recent findings have confirmed the utility of intradermal therapy in managing the subjective symptoms and clinical signs of PEFS [91,92]. As proposed in two different consensus statements [1,15], it has thus been suggested that CVD should be managed with standard systemic treatments combined with local mesotherapy, specifically applied to areas affected by microcirculatory alterations, to improve microcirculation and reduce both symptoms and functional limitations. While awaiting further data to assess the medium- and long-term efficacy, the management of CVD and PEFS may potentially benefit from mesotherapy. However, it should be applied based on an algorithm tailored to the clinical and instrumental response of the individual patient (Figure 4).

Weekly mesotherapy sessions may become biweekly and eventually monthly once results are achieved and documented through objective and instrumental parameters, and a regular weekly follow-up is necessary to evaluate both the efficacy and tolerability of the treatment. Mesotherapy should be considered as an add-on to the systemic treatment of chronic venous insufficiency in the lower limbs (Table 12).

### 3.10. Question 10. Can Mesotherapy Be Considered in Dermatology?

Mesotherapy is a particularly interesting technique in dermatology because the target organ is directly the skin and the lesion to be treated. Certain skin reactions induced by the mesotherapy technique, collectively referred to as mesodermal modulation, suggest an enhancement of local pharmacological action when a product is deposited in the dermis (e.g., increased primary and secondary immune response, analgesic effects induced by microtrauma, and amplification of the injected drug’s mechanism of action) [8,93,94]. In 2020, after reviewing the various dermatological applications of mesotherapy (Table 13), the authors concluded the following: “The dermis appears to be a target organ for many pathologies and clinical dermatologists will play a crucial role in the development of mesotherapy.” “Mesotherapy applied in dermatology needs more clinical confirmation. Studies are needed to identify the ideal drug concentration, the depth of infiltration, and the patients who benefit most from this technique” [93]. In clinical practice, certain conditions seem to benefit more from mesotherapy, such as hair loss [95], alopecia [96,97], and melasma management [98,99] (Table 14). However, it should be noted that, in some dermatologic conditions, treatment has primarily been intra-lesional, such as in acne [100], keloids [101], warts [102], and others. In dermatology, not all studied agents have been approved by regulatory authorities for the indications under investigation. The off-label use of such products is therefore subject to national regulations in each country.

### 3.11. Question 11: Can Mesotherapy Be Proposed for Managing Skin Aging?

The term skin aging refers to various alterations resulting from changes in connective tissue and fibroblasts, leading to the loss of elasticity, wrinkles, telangiectasias, and actinic keratosis [103,104]. Some medical devices have been used intradermally with the goal of promoting hydration, proliferation, migration, and cellular reorganization, and to prevent chrono- and photoaging [105]. Some data suggest improvements in skin appearance and quality (reduction in fine wrinkles, increased hydration, turgor, elasticity, and firmness) [106,107,108]. However, due to the lack of histological results, it is not currently possible to draw conclusions about the long-term efficacy and tolerability of these products [109]. Autologous platelet-rich plasma (PRP) has also provided some positive data [110], but these procedures have not received favorable opinions from some national authorities [111,112]. In esthetic medicine, product mixtures have been used in the belief they would yield better results. However, a recent review has reported that individual products may be more effective than mixtures [113]. Even in the dermo-esthetic field, therapeutic decisions should therefore be based on a risk/benefit assessment according to the clinical diagnosis, considering ethical and clinical research findings free from conflicts of interest [114] (Table 15).

### 3.12. Question 12: How Can Adverse Events Reported in the Literature Be Prevented?

Like any other technique, mesotherapy may cause three types of adverse events:Events caused by the microtrauma produced by the needle (e.g., mild pain at the injection site and bruising) [115];Those caused by the product or mixtures injected (e.g., local and systemic reactions) [116,117,118];

Those caused by noncompliance with hygiene standards or malpractice (e.g., improper execution, wrong drug selection, lack of sterilization of the environment or skin, and failure to use sterile disposable equipment) [119,120,121].

Over the years, various substances have been used, including allopathic, homeopathic, herbal products, and medical devices, often in combination, particularly for dermo-esthetic purposes. However, it should be noted that not all products used are approved for such purposes or for intradermal/subcutaneous use, so any adverse event should be recorded and reported to the appropriate regulatory authorities, even in the case of treatments performed for esthetic purposes [19] (Table 16).

### 3.13. Question 13: Can Mesotherapy Technique Be Applied to the Oral Mucosa?

Mesotherapy has also been used for the management of gingival hyperpigmentation [122,123,124,125]. However, due to the profound anatomical and functional differences between skin and oral mucosa, the use of the term “mesotherapy” in dentistry is not straightforward: many of the mechanisms of action hypothesized for mesotherapy on the skin may not function in the same way on oral mucosa. The infiltration technique, when applied to the oral mucosa, appears promising based on currently available data, but only future RCTs will facilitate a better understanding of the role of this infiltration technique in the oral region (Table 17).

### 3.14. Questions 14 and 15: Can Mesotherapy Be Part of a Multimodal Treatment Strategy? What Treatment Algorithm Is Recommended for the Application of Mesotherapy?

Mesotherapy has been used in combination with TENS, laser therapy, rehabilitation exercises, and various systemic analgesic treatments [23,69,126]. A synergistic effect has also been described in dermatology for the treatment of melasma [98,127], the management of alopecia [95], and acne [128]. The benefit of combining two different therapeutic strategies is enhanced by mesotherapy, since it uses lower drug dosages and reduces recovery times [3,23,26,129]. Mesotherapy may therefore be considered in the therapeutic pathway for one of the most sought-after advantages by practicing physicians: the reduction in potential drug interactions and the use of the minimum effective dose. As a result, the combination of a local treatment like mesotherapy with a systemic treatment, appropriately tailored to the patient’s needs, can represent a recommended strategy, especially in patients at risk of drug interactions (such as the elderly, poly-treated patients, or those with chronic conditions) [130]. In the management of chronic musculoskeletal pain, mesotherapy may be synergistic with other therapeutic strategies, as it not only reduces the systemic dose of analgesics but also improves treatment acceptance. To this end, an algorithm has been proposed to guide the treatment pathway for individual patients, allowing combined treatments with mesotherapy and other therapeutic strategies to be considered (Figure 5). Possible future interactions with other complementary techniques, such as acupuncture, should be studied.

### 3.15. Question 16: When Can Mesotherapy Be Applied in Clinical Practice?

Mesotherapy is a technique that can be used for preventive, curative, or rehabilitative purposes in an individualized care pathway, in association with other therapies, either pharmacological or non-pharmacological, or on its own when other proven options have failed or cannot be used, or when no other therapeutic options are available. This is particularly applicable to patients who have undergone a medical examination from which there is a rationale for localized treatment [1]. The choice of the product to be injected should be based on its tolerability and efficacy, which justify its use, and should only be administered after obtaining valid informed consent [3] (Table 18).

### 3.16. Question 17: Are There Clinical Conditions That Contraindicate Mesotherapy?

The intracutaneous administration technique involves a microtrauma induced by the needle. Conditions that cause significant coagulation defects or the use of anticoagulant medications may therefore lead to prolonged bleeding and the formation of hematomas resulting from the needle insertion [1]. The presence of autoimmune diseases, a tendency to develop hypertrophic or keloid scars, active herpes infections, ongoing therapies that can modify cutaneous repair processes, the presence of permanent fillers at the injection sites, and allergic diathesis are contraindications for mesotherapy. These should therefore be investigated in the patient’s medical history to assess the risk/benefit ratio of mesotherapy. The injection of a drug involves systemic absorption, so the contraindications of the injected drugs should also be considered [3]. These and other ethical considerations are listed in Table 19**.**

### 3.17. Question 18: Is a Specific Informed Consent Required for Mesotherapy?

Only after being properly informed can the patient knowingly participate in the treatment pathway and more accurately assess the risk/benefit ratio and the results achievable. In the doctor–patient relationship, with the support of informed consent, the patient is encouraged to evaluate the risk/benefit ratio throughout the entire treatment process [131]. As with any other treatment, the patient must also be informed about the various therapeutic options available and the risks of each. The patient must be properly informed, even when using active ingredients not approved for intradermal or subcutaneous routes; off-label use must also be shared with the patient [132]. It is therefore advisable to keep a written record of the consent obtained (Table 19).

### 3.18. Question 19: Is It Necessary to Report the Effects of Mesotherapy in the Patient’s Medical Chart?

Recording clinical, diagnostic, and therapeutic parameters is considered good clinical practice [133]. Completing the medical chart (whether paper or digital) holds ethical and medico-legal value and facilitates the retrospective review of clinical data through audits [134]. For example, in Italy, according to Law No. 38 of 15 March 2010, physicians are required to complete the medical chart, periodically documenting pain levels and results obtained from treatments [1] (Table 19).

### 3.19. Question 20. Can Mesotherapy Be Performed on Minors?

Many active ingredients have not been approved for use in pediatric patients [135]. Mesotherapy in pediatrics is not routine, and limited data in this age group are available [136,137,138,139]. Additionally, skin thickness in pediatric patients is significantly different from adults [139], making it challenging to apply results obtained in adults to pediatric patients. However, in pediatric patients, the micro-needle-based drug delivery technique has been considered to bypass the oral route for medications that pediatric patients cannot swallow [140,141]. Considering its analgesic effect, needle puncture alone may be a viable option in pediatric patients, depending on the pain it elicits. In fact, some data seem to encourage research on acupuncture in certain clinical situations in pediatrics [142] (Table 19).

### 3.20. Question 21: Who Can Practice Mesotherapy

Mesotherapy is a medical procedure that must be performed exclusively by qualified and licensed physicians. As it involves the injection of pharmacological agents into the skin, it constitutes a clinical act that requires full medical responsibility, including diagnosis, treatment planning, administration, and follow-up. These responsibilities carry medical, ethical, and legal implications and cannot be delegated to non-medical personnel. Worldwide, the practice of mesotherapy should fall within the professional scope of physicians—both specialists and general practitioners—who have received specific training and demonstrated clinical competence in this technique. Its application in patient care must be conducted under medical supervision to ensure safety, therapeutic effectiveness, and adherence to current scientific evidence and clinical guidelines. General practitioners, particularly those managing pain, dermatological conditions, and chronic diseases, may implement mesotherapy effectively when properly trained. Ongoing education and strict compliance with evidence-based protocols are essential to maintain high standards of care and to minimize the risk of inappropriate use. This statement proposes a universal framework for training and practice, supporting the safe and effective integration of mesotherapy into evidence-based medical care across diverse healthcare systems.

### 3.21. Question 22: What Is the Role of Research?

The evidence supporting mesotherapy as a route of administration is abundant, as demonstrated by the increase in scientific publications on the topic. However, many aspects remain unclear, and rigorous studies are still needed. Only through this approach will it be possible to complete the transition towards evidence-based mesotherapy [8]. The priority research areas are summarized in Table 20.

Some preclinical animal studies have confirmed that mesotherapy induces cell-mediated actions in the dermis, which enhance the local pharmacological action [1,143,144]. The dermis may therefore represent the target for new local and systemic therapies [93]. It is currently believed that more superficial inoculation, within a depth of 2 mm, facilitates a slower diffusion of the drug into the underlying tissues. However, further studies are needed to precisely identify the correct inoculation depth. Finally, it is urgent to study the medium- and long-term effects of dermo-esthetic products introduced to the market. In this area of application, the benefit of each product should be compared with placebos, using validated scales. This would facilitate transparent communication with the patient regarding the real possibilities of the injected product. In addition to preclinical and clinical research, it is particularly useful to identify the potential economic benefit of mesotherapy through Health Technology Assessment (HTA) methodologies. Indeed, the economic impact of mesotherapy, when applied according to standard criteria, has been positive [145]. Calibrated audits, based on patient needs, will also help identify the best care pathways (Table 21).

### 3.22. Question 23: What Do Patients Recommend?

The involvement of patient associations has helped identify several areas of discussion, leading to the development of specific recommendations (Table 22). The time spent to fully align the treatment pathway was considered a fundamental part of the medical consultation and essential to obtaining valid informed consent [131]. It is becoming clear that the doctor’s experience alone is not sufficient to ensure the safety of a particular treatment, and evidence of tolerability and efficacy is necessary. This issue carries significant medico-legal implications, particularly in the context of mesotherapy procedures supported by insufficient or emerging evidence.

## 4. Potential Impact of the Guideline on Care Pathways

Mesotherapy, in some of its applications, meets the criteria of essential care, avoiding enormous costs and improving the management of certain symptoms. In fact, a pharmacoeconomic analysis conducted to evaluate the resource absorption in patients starting treatment with systemic NSAIDs showed a saving for the Healthcare System generated by locoregional infiltrations [145]. If mesotherapy were integrated into LEA, certain populations would benefit from it, for example, patients with chronic pain, elderly patients, and those on polytherapy at high risk for pharmacological interactions. In these patient subgroups, localized treatment could lead to clinical benefits and an improvement in the quality of life, with significant savings in medications. Obviously, the economic impact of mesotherapy, due to its drug-sparing effect, should be measured in each country, where costs and healthcare pathways may differ significantly. For the same clinical effect, patient satisfaction and the reduced time spent by healthcare providers using mesotherapy should also be considered.

## 5. Emerging Recommendations

It is 50 years since the Italian Society of Mesotherapy was founded, and the first consensus on its fundamental principles has been reached. This document emphasizes the importance of distinguishing between superficial (intradermal) mesotherapy and deeper injection techniques. It excludes procedures such as “scarification” from the definition of mesotherapy and proposes dry needling as a comparator to evaluate the effects of micro-punctures versus micro-injections.

Key procedural variables—such as needle type, insertion angle, injection depth, the number and spacing of micro-injections, dose per site, and total volume injected—are now recognized as critical to understanding mesotherapy’s mechanisms of action and assessing its clinical benefits. While its primary effect is local and pharmacological, additional immune and systemic interactions contribute to a drug-sparing effect when compared to systemic administration. 

These insights broaden the clinical relevance of mesotherapy in pain management, rehabilitation, and sports medicine. In the latter, the potential systemic absorption of injected substances raises concerns regarding anti-doping compliance, making drug selection—based on diagnosis, tolerability, efficacy, and regulatory status—critically important. Mesotherapy should never be guided by personal beliefs, but by a sound clinical rationale.

Mesotherapy represents a valuable therapeutic option, especially for patients requiring reduced drug exposure—such as the elderly, polymedicated individuals, and those at high risk of drug interactions or intolerance to conventional therapies. Whether used alone or as an adjunct, it may improve treatment adherence. In phlebology and dermatology, it is an established adjuvant technique, with procedural choices guided by diagnosis and the characteristics of available products. In esthetic dermatology, its application must be grounded in demonstrated efficacy and safety.

Crucially, mesotherapy must be preceded by a thorough medical examination to rule out contraindications and establish a clear therapeutic rationale. Strict adherence to aseptic techniques is mandatory, as any deviation may constitute malpractice. As with all infiltrative procedures, mesotherapy requires a careful benefit/risk assessment involving the patient. Ethical and legal responsibilities include comprehensive documentation, obtaining valid informed consent, and ensuring that the procedure is performed exclusively by qualified medical professionals. While these practices may seem self-evident, they are fundamental to guaranteeing safe and appropriate care. Notably, “oral mesotherapy” was the only topic that failed to achieve 70% expert consensus, with 34% of participants abstaining—underscoring the need for further investigation in this area. Nevertheless, preliminary studies suggest that mucosal tissues may also benefit from infiltration using mesotherapy techniques.

No significant differences emerged between national and international expert opinions, reinforcing the global applicability of the recommendations. Although some may view them as either overly stringent or too general, the rigorous methodology employed (7)—rooted in expert consensus and existing literature—supports the reliability of these evidence-based standards in guiding both clinical and policy decisions. Mesotherapy can now be justifiably integrated into prevention, treatment, and rehabilitation pathways.

## 6. Limitations of This Document

Despite the number of studies examined, the quality of some of them did not allow us to draw strong conclusions. However, the broad multidisciplinary consensus and methodological rigor have allowed this limitation to be partially overcome.

## 7. Conclusions

Mesotherapy has the potential to be integrated into routine clinical practice, particularly for selected patient groups—such as older adults, individuals with musculoskeletal pain, chronic venous insufficiency, and certain dermatologic conditions. The failure or low efficacy of traditional therapies prompts the inclusion of complementary techniques in patient care processes, especially in pain medicine. Scientific societies and health authorities are encouraged to address current gaps in evidence and assess the risks linked to the off-label use of mesotherapy products, especially regarding safety and ethical implications. Advancing the field will require high-quality research—particularly randomized controlled trials—that can establish mesotherapy as a standardized evidence-based intervention. This goal could be supported through targeted public and private investment, thus helping to avoid redundant or low-impact studies.

## Figures and Tables

**Figure 1 jcm-14-04689-f001:**
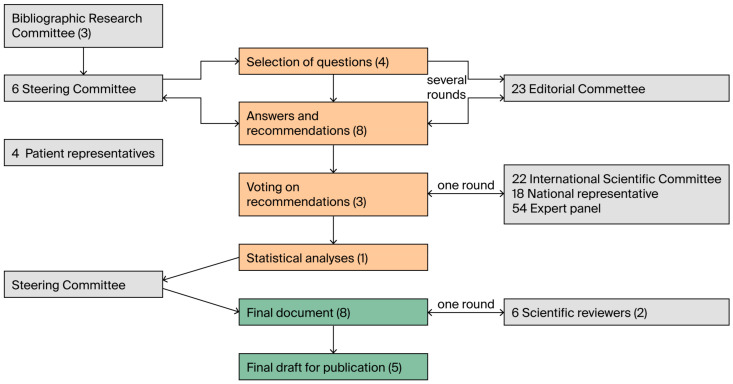
This figure illustrates the steps of the entire scientific process. The literature review included studies published up to December 2023. The drafting of the final document was completed in October 2024, and the final draft of the publication in April 2025. The number of experts involved in each committee is reported, while the duration in months for each phase is shown in parentheses.

**Figure 2 jcm-14-04689-f002:**
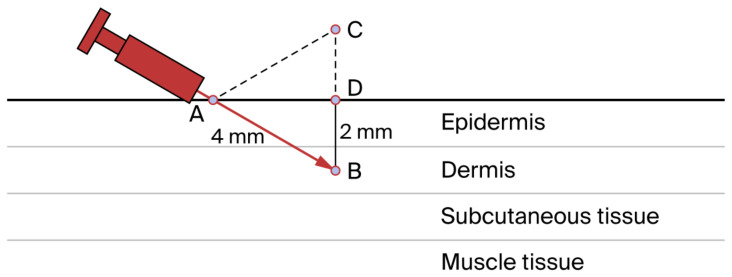
The figure schematically shows the skin layers and the 4 mm needle (in red). The 4 mm needle length represents the side of an equilateral triangle (dashed lines). AB represents the needle with a length of 4 mm. AB inserted with an inclination of 30° constitutes one side of an equilateral triangle (ABC). AB = 4 mm; DB = 2 mm. By increasing or decreasing the angle, different injection depths can be achieved. Note that in the figure, the 4 mm needle length represents the side of an equilateral triangle (dashed lines).

**Figure 3 jcm-14-04689-f003:**
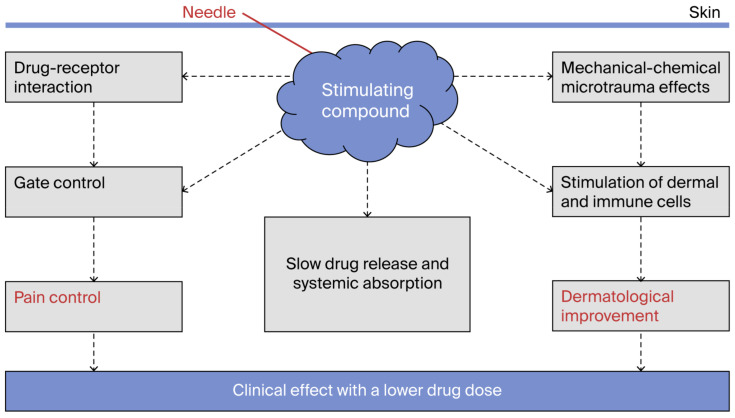
Mesodermal modulation. The figure schematically shows the possible mechanisms of action. The drug triggers local reactions and is slowly absorbed systemically. In the dermis, the distension of the tissues induced by the needle and the injected drug contribute to the analgesic effect.

**Figure 4 jcm-14-04689-f004:**
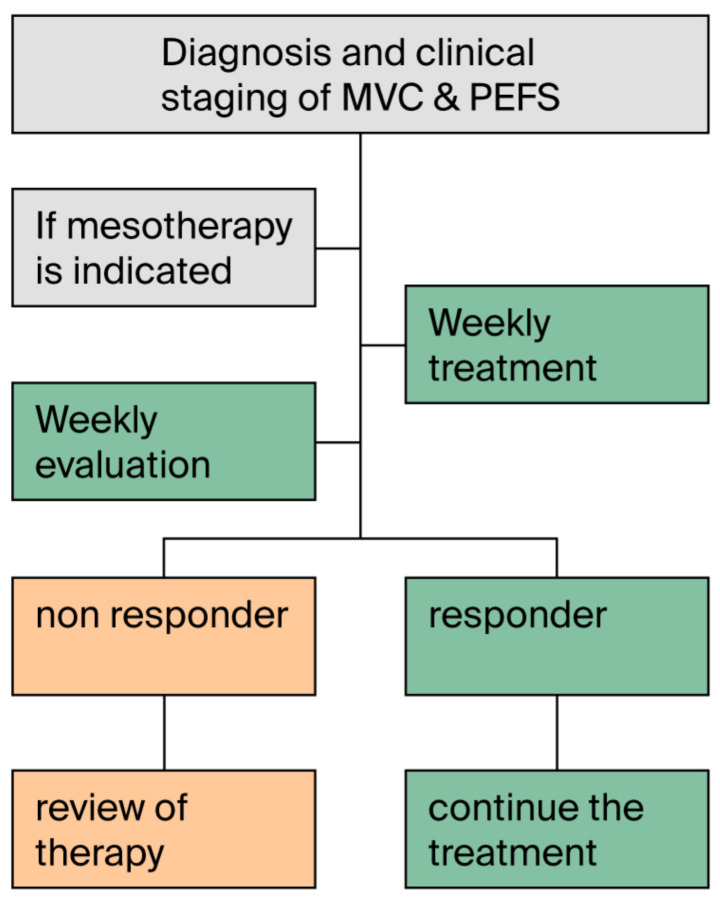
The figure shows a decision-making algorithm for the application of mesotherapy in chronic venous disease (CVD) and edematous-fibrosclerotic panniculopathy (PEFS) based on disease staging and treatment outcomes.

**Figure 5 jcm-14-04689-f005:**
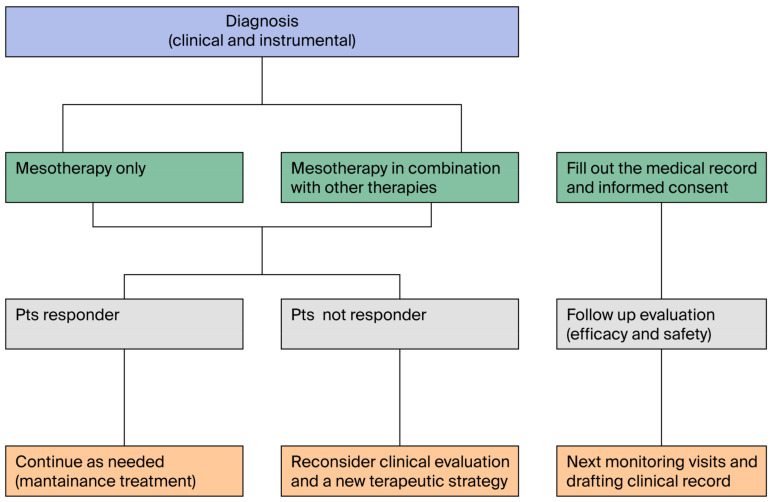
The figure shows the treatment algorithm for mesotherapy alone or in combination with other therapeutic strategies.

**Table 1 jcm-14-04689-t001:** Key Questions.

N	Questions
1	How should mesotherapy be defined?
2	How is mesotherapy performed?
3	What is the mechanism of action of mesotherapy?
4	What is the advantage of the drug-sparing effect of the intradermal route in immunoprophylaxis?
5	Which substances are injected?
6	Can mesotherapy be included in the management of patients with localized musculoskeletal pain?
7	Can mesotherapy be integrated into the Individual Rehabilitation Plan (IRP)?
8	Can sports trauma benefit from mesotherapy?
9	Can mesotherapy be included in the care pathway for the signs and symptoms of chronic venous disease (CVD) and the prevention of its complications (PEFS)?
10	Can mesotherapy be considered in dermatology?
11	Can the mesotherapy technique be proposed for the management of skin aging?
12	How can the adverse events reported in the literature be prevented?
13	Can the mesotherapy technique be applied to the oral mucosa?
14	Can mesotherapy be part of a multimodal treatment strategy?
15	What treatment algorithm is recommended for the application of mesotherapy?
16	When can mesotherapy be applied in clinical practice?
17	Are there any clinical conditions that contraindicate mesotherapy?
18	Is a specific informed consent required for mesotherapy?
19	Is it necessary to report the effects of mesotherapy in the patient’s medical chart?
20	Can mesotherapy be performed on minors?
21	Who can practice mesotherapy?
22	What is the role of research?
23	What do patients recommend?

Listed here are the 23 questions aimed at standardizing the mesotherapy technique. The Scientific Committee used the PICO method to structure these 23 questions. For P (Population), the target population/patients were those who would most benefit from the minimum effective drug dose; for I (Intervention), any clinical condition or disease responsive to localized treatment was considered; for C (Comparison), each clinical question included studies comparing or associating other methods with mesotherapy; for O (Outcome), tolerability, safety, and effectiveness of the mesotherapy treatment were evaluated.

**Table 2 jcm-14-04689-t002:** Levels of evidence by the US Agency for Healthcare Research and Quality.

Level of Evidence	Requirements
Ia	Evidence from meta-analysis of randomized trials
Ib	Evidence obtained from at least one RCT
IIa	Evidence obtained from at least one well-designed controlled trial without randomization
IIb	Evidence obtained from at least one other well-designed experimental study
III	Evidence obtained from well-designed non-experimental descriptive studies, such as comparative, correlational and case studies
IV	Evidence obtained from expert reports or authoritative opinions and/or clinical experiences

**Table 3 jcm-14-04689-t003:** Result of the experts’ vote.

No.	Definition of Mesotherapy	Rif Question	AHCPR	Mean ± SD	%		
					Agreement	Abstained	Disagreement
1	The term “mesotherapy” describes the technique with which microinjections are performed into the thickness of the skin for preventive, curative or rehabilitative purposes	Q1	IV	4.8 ± 0.6	96.6%	1.7%	1.7%
2	The term “local intradermal therapy” describes the technique with which a series of microinjections are performed in the superficial dermis of a specific skin area	Q1	IV	4.8 ± 0.5	97.5%	1.7%	0.8%

**Table 4 jcm-14-04689-t004:** Result of the experts’ vote.

No.	Administration Technique	Rif Question	AHCPR	Mean ± SD	%		
					Agreement	Abstained	Disagreement
3	Mesotherapy is performed with a 4mm (27 Gauge) or 13mm (30 G - 32 G) needle. Depending on the length of the needle and the thickness of the skin to be treated, the angle of inclination and the insertion depth of the needle itself will vary. The 4 mm needle inclined at 30° to the skin surface allows inoculation at approximately 2 mm depth. By increasing or reducing the inclination angle, different inoculation depths are obtained	Q2	IV	4.7 ± 0.7	94.1%	3.4%	2.5%
4	To carry out local intradermal therapy, the formation of micro drug deposits in the dermis (wheals) is recommended, to obtain which 0.1-0.2 ml of liquid must be inoculated for each single microinjection	Q2, Q3	IV	4.7 ± 0.7	94.1%	3.4%	2.5%
5	The distance between one microinjection and another varies from 1 to 2 cm	Q2	IV	4.5 ± 0.9	86.6%	9.2%	4.2%
6	Mesotherapy must be performed strictly observing the rules of asepsis	Q2	IV	4.9 ± 0.3	99.2%	0.8%	0.0%
7	If the use of two active ingredients is necessary, the administration of the individual products in different syringes and in different inoculation sites is recommended	Q2	IV	4.4 ± 1	83.2%	10.1%	6.7%
8	Multi-injectors are not recommended	Q2	IV	4.5 ± 0.9	83.2%	10.90%	5.9%

**Table 5 jcm-14-04689-t005:** Result of the experts’ vote.

No.	Mechanism of Action	Rif Question	AHCPR	Mean ± SD	%		
					Agreement	Abstained	Disagreement
9	The effect of mesotherapy depends on the predominantly local action of the injected drug to which systemic absorption, reactions induced by the needle, tissue distension caused by the liquid, cell-mediated and neuro- immune reactions can contribute. The set of these mechanisms is defined as “mesodermal modulation”	Q3	IV	4.8 ± 0.5	95.8%	3.4%	0.8%
10	Intradermal administration produces a series of wheals that constitute a “reserve” from which the drug is slowly absorbed with the aim of prolonging its effect	Q2, Q3	IIb	4.8 ± 0.6	97.5%	1.7%	0.8%
11	Mesotherapy allows a drug-sparing effect and an efficacy comparable to that of systemic therapy	Q3, Q4	IIa	4.5 ± 0.8	88.2%	10.9%	0.8%

**Table 6 jcm-14-04689-t006:** Result of the experts’ vote.

No.	Drug-Sparing Effects on Immunoprophylaxis	Rif Question	AHCPR	Mean ± SD	%		
					Agreement	Abstained	Disagreement
12	The intradermal route induces an antibody response equal to or greater than the intramuscular route, but with a lower dose of antigen	Q4	Ib	4.5 ± 0.7	90.8%	9.2%	0.0%

**Table 7 jcm-14-04689-t007:** Result of the experts’ vote.

No.	Pharmacology	Rif Question	AHCPR	Mean ± SD	%		
					Agreement	Abstained	Disagreement
13	To apply mesotherapy, the use of active ingredients indicated in the pathology or symptom to be treated is recommended	Q5	IV	4.8 ± 0.5	95.8%	3.4%	0.8%
14	In mesotherapy it is recommended to use injectable products and to consider those not indicated for the mesotherapy route as off-label	Q5, Q12	IV	4.7 ± 0.6	95.0%	4.2%	0.8%
15	The use of mixtures is permitted only if the products have authorization for use in combination or if they have efficacy and tolerability studies	Q5, Q12	III	4.7 ± 0.7	93.3%	5.0%	1.7%

**Table 8 jcm-14-04689-t008:** Studies on localized pain.

Reference	Disease	Number of Patients	Comparison	Follow-Up	N of Sessions	Outcome
[60]	Cervico brachialgia	20	TENS	20 days	6	Improvement and reduced need for therapy
[56]	Acute lumbosciatica	44	Placebo	1 day	1	Good efficacy and tolerability
[57]	Low back pain	22	Laser	1 year	8	Better results with mesotherapy
[58]	Low back pain	84	Systemic therapy	6 months	5	Same effect as systemic therapy
[37]	Low back pain	62	Trigger points	12 weeks	4	Better results of mesotherapy on trigger points
[59]	Calcific tendinitis of the shoulder	80	Placebo	1 year	3	Reduction of calcifications
[64]	Acute low back pain	68	Placebo	1 day	1	Improved pain, mobility, and quality of life
[65]	Osteoarthritis	50	Oral therapy	6 months	3	Improved pain and functionality
[42]	Low back pain	168	Placebo	1 day	1	Improved pain and quality of life
[66]	Chronic neck pain	42	Placebo	3 months	3	Improved pain and quality of life
[67]	Acute neck pain	36	Oral therapy	3 days	1	Improved pain and quality of life
[68]	Osteoarthritis	117	Oral therapy	3 months	9	Improved pain and mobility
[39]	Chronic spinal pain	217	Placebo	3 months	5	Improved pain and mobility
[69]	Fibromyalgia-related neck pain	78	Placebo	3 months	7	Improved pain and functionality
[25]	Osteoarticular pain	141	1 drug vs. 2 drugs	3 months	9	Improved pain and reduced drug consumption
[24]	Low back pain in emergency dept	120	Intravenous therapy	1 day	1	Mesotherapy superiority and reduced drug need
[27]	Acute musculoskeletal injuries in emergency dept	96	Intravenous therapy	1 day	1	Mesotherapy superiority and reduced drug need

**Table 9 jcm-14-04689-t009:** Result of the experts’ vote.

No.	Localized Pain	Rif Question	AHCPR	Mean ± SD	%		
					Agreement	Abstained	Disagreement
16	Mesotherapy represents an option in the management of localized musculoskeletal pain	Q6	Ia	4.9 ± 0.4	98.3%	1.7%	0.0%
17	Mesotherapy is recommended in the management of localized pain when the drug-sparing effect and the potential lower systemic pharmacological impact represent an advantage	Q6, Q7, Q8	Ia	4.8 ± 0.5	96.6%	3.4%	0.0%
18	It is recommended to determine the frequency, number of sessions and duration of treatment based on the clinical response. The available studies report a frequency of sessions usually weekly with a number of sessions from 1 to 9	Q6, Q7, Q8	Ib	4.7 ± 0.5	97.5%	2.5%	0.0%

**Table 10 jcm-14-04689-t010:** Result of the experts’ vote.

No.	Rehabilitation	Rif Question	AHCPR	Mean ± SD	%		
					Agreement	Abstained	Disagreement
19	Mesotherapy is applicable in individual rehabilitation programs	Q6, Q7	Ib	4.7 ± 0.6	95.0%	5.0%	0.0%
20	Mesotherapy is applicable in Sports and Exercise Medicine	Q6, Q7, Q8	IIb	4.8 ± 0.6	93.3%	6.7%	0.0%
21	Mesotherapy applied in Sports and Exercise Medicine must take anti- doping regulations into account	Q8	IV	4.7 ± 0.6	90.8%	9.2%	0.0%

**Table 11 jcm-14-04689-t011:** Studies on sports injuries.

Reference	Disease	N of Pts	Follow-Up	Number of Sessions	Outcome
[75]	post traumatic pain	133	4 months	1–4 sessions	Positive efficacy/safety; functional recovery of sporting competitive activity in shorter time than conventional therapies
[76]	pubic myoenthesitis	256	6 months	from 2 to 5 sessions at 10–20 days intervals	Complete functional recovery after 4 sessions
[77]	acute lumbo sciatic pain in athletes	20	4 months	2–6 sessions	Pain reduction and functional recovery in 90% of patients
[78]	patellar tendonitis	126	1 month	weekly sessions	85% of patients reach complete pain relief (from 1 to 4 sessions)
[79]	ileo-tibial band friction syndrome	40	3 months	weekly sessions	Pain relief in 55% of patients after 2 sessions; 97.5% after 3 sessions
[80]	myoenthesitis of the leg	203	2 months	sessions at 7–8 days intervals	60.8% of patients reach complete recovery with 1 session; 96,6% of patients reach complete recovery with 3 sessions. Mesotherapy was more effective for patients with recent pain.
[81]	rotator cuff tendinopathy	145	12 weeks	from 4 to 9	Reduction in pain, improvement in functioning
[82]	achilles tendonitis	40	12 weeks	4 weekly sessions	Pain reduction

**Table 12 jcm-14-04689-t012:** Results of expert votes.

No.	Chronic Venous Disease and Its Complications	Rif Question	AHCPR	Mean ± SD	%		
					Agreement	Abstained	Disagreement
22	Mesotherapy is applicable in Chronic Venous Disease for the management of signs and symptoms, to limit its evolution and prevent complications	Q9	III	4.5 ± 0.7	84.9%	15.1%	0.0%
23	Mesotherapy is applicable in the management of fibro-sclerotic edematous panniculopathy (PEFS)	Q9, Q12	IIb	4.5 ± 0.8	81.5%	18.5%	0.0%

**Table 13 jcm-14-04689-t013:** Dermatological conditions in which mesotherapy has been applied.

Dermatological Disorder
Alopecia
Cystic acne
Keloid
Cyst suppurated
Suppurative hydrosadenitis
Psoriasis
Ring granuloma
Foreign body granuloma
Lichen planus
Neurodermatitis and prurigo
Postscabular nodules
Warts
Benign lymphocytic infiltration
Cutaneous or discoid lupus
Lupic panniculitis
Cutaneous leishmaniasis
Eczema
Vitiligo
Lipoid necrobiosis
Pretibial myxedema
Cutaneous neoplasms

**Table 14 jcm-14-04689-t014:** Result of the experts’ vote.

No.	Dermatology	Rif Question	AHCPR	Mean ± SD	%		
					Agreement	Abstained	Disagreement
24	Mesotherapy is applicable in the management of some dermatological conditions	Q10, Q12	III	4.5 ± 0.8	84.0%	16.0%	0.0%
25	Mesotherapy represents an option in the treatment of alopecia	Q10	Ia	4.4 ± 0.9	78.2%	20.2%	1.7%
26	Mesotherapy represents an alternative or combination therapy in the treatment of melasma in patients refractory to first line therapy	Q10	Ia	4.0 ± 1.0	62.2%	36.1%	1.7%

**Table 15 jcm-14-04689-t015:** Result of the experts’ vote.

No.	Mesotherapy in Skin Aging	Rif Question	AHCPR	Mean ± SD	%		
					Agreement	Abstained	Disagreement
27	The blemish must be framed from a medical point of view in order to identify the rationale for treatment with mesotherapy	Q11, Q12	IV	4.5 ± 0.8	81.5%	18.5%	0.0%
28	Mesotherapy can be considered in the management of some blemishes if the goal of treatment is rational and if the patient shares the risk/benefit	Q11, Q12	IV	4.5 ± 0.8	82.4%	16.8%	0.8%

**Table 16 jcm-14-04689-t016:** Result of the experts’ vote.

No.	Contraindications and Risk Management	Rif Question	AHCPR	Mean ± SD	%		
					Agreement	Abstained	Disagreement
29	Mesotherapy must be performed by medical personnel and cannot be delegated to another healthcare professional	Q12, Q18	IV	4.9 ± 0.3	98.3%	1.7%	0.0%
30	Mesotherapy must be performed in patients who have undergone a medical examination from whicha rationale in favor of this treatment has emerged	Q12, Q18	IV	5.0 ± 0.2	100.0%	0.0%	0.0%
31	Mesotherapy must not be applied in subjects with absolute contraindications due to the technique or the injected product	Q17	IV	5.0 ± 0.2	99.2%	0.8%	0.0%
32	Mesotherapy must be performed in a suitable environment to guarantee asepsis and infection prevention standards	Q12, Q18	IV	5.0 ± 02	100.0%	0.0%	0.0%
33	Every adverse event must be recorded in the patient’s medical record and communicated to the health authorities according to current regulations	Q19	IV	5.0 ± 0.3	98.3%	1.7%	0.0%

**Table 17 jcm-14-04689-t017:** Result of the experts’ vote.

No.	Scientific Research	Rif Question	AHCPR	Mean ± SD	%		
					Agreement	Abstained	Disagreement
34	The use of the superficial infiltration technique applied to the oral mucosa (known as “oral mesotherapy”) has yielded promising data, but while awaiting further studies, the patients should be informed that its clinical application is experimental	Q13, Q22	IV	4.1 ± 1.0	63.9%	33.6%	2.5%

**Table 18 jcm-14-04689-t018:** Result of the experts’ vote.

No.	Combination with Other Treatment Strategies and Treatment Algorithms	Rif Question	AHCPR	Mean ± SD	%		
					Agreement	Abstained	Disagreement
35	Mesotherapy is applicable in the treatment path of patients and also in combination with other treatments	Q14	III	4.9 ± 0.3	100.0%	0.0%	0.0%
36	The treatment algorithm, in each area of application of mesotherapy, must consider the clinical response	Q15	IIb	4.9 ± 0.3	99.2%	0.8%	0.0
37	Mesotherapy must be applied in a personalized treatment path, after a diagnosis and an accurate pharmacological, allergy, and pathological history	Q16; Q18, Q12	IV	5.0 ± 0.2	100.0%	0.0%	0.0%
38	Mesotherapy can be suggested exclusively to patients who have undergone a medical examination from which a rationale in favor of localized treatment has emerged	Q16, Q18	IV	4.9 ± 0.4	98.3%	0.8%	0.8%
39	In the individualized treatment path, mesotherapy can be used in combination with other therapies, pharmacological or non-pharmacological, or alone when other options with proven efficacy have failed or cannot be used, or there are no other therapeutic options	Q16; Q3	IV	4.8 ± 0.5	97.5%	2.5%	0.0%

**Table 19 jcm-14-04689-t019:** Result of the experts’ vote.

No.	Ethics	Rif Question	AHCPR	Mean ± SD	%		
					Agreement	Abstained	Disagreement
40	Before introducing mesotherapy into the individual treatment path, the doctor must explain its advantages and limitations, specify the product or products used, and obtain written informed consent	Q18	IV	5.0 ± 0.2	99.2%	0.8%	0.0%
41	Information documents provided to the patient to obtain informed consent must be based on current guidelines	Q18	IV	4.9 ± 0.5	96.6%	2.5%	0.8%
42	It is recommended to fill in the clinical record with the diagnosis, products used and their quantity injected, number of sessions, and results obtained	Q19	IV	5.0 ± 0.1	100.0%	0.0%	0.0%
43	Mesotherapy must be considered like any other off-label therapy even in minor patients, when the product, the route of administration, and the age of the patient do not fall within the authorization of the injected drug	Q20	IV	4.7 ± 0.6	92.4%	6.7%	0.8%
44	The teaching and updating of mesotherapy must be based on current guidelines	Q21	IV	4.9 ± 0.3	99.2%	0.0%	0.8%

**Table 20 jcm-14-04689-t020:** The table shows the main areas of investigation.

**Preclinical Research**
1. Dose (tissue) effect curve of the drug injected via ID
2. Role of the dermis (dermal cells) in the clinical response
**Clinical Research**
1. Injection depth
2. Comparison of drug vs. combination of drugs
3. Pharmacodynamic differences between ID and IV routes
4. Cost/benefit in the various application areas (pain, MVC, dermo-aesthetic)
5. Role of the ID pathway in vaccination
**Heath Technology Assessment**
1. Economic impact of mesotherapy compared to other therapies
2. Quality of life
3. Patient acceptability
4. Efficiency for the Healthcare System Clinical-organizational
**Audit**
1. Efficiency in diagnostic, therapeutic and healthcare pathways
2. Update of the guidelines

**Table 21 jcm-14-04689-t021:** Result of the experts’ vote.

No.	Scientific research	Rif Question	AHCPR	Mean ± SD	%		
					Agreement	Abstained	Disagreement
45	Researchers are recommended to draw up protocols useful for a better understanding of the mechanism of action and the role of mesotherapy in treatment pathways	Q22	IV	4.9 ± 0.3	99.2%	0.8%	0.0%
46	Clinicians are recommended to publish data relating to mesotherapy with a description of the technique used (depth of injection, number of micro-injections, treatment area, quantity of drug injected, number and frequency of sessions) and use methods of collecting results according to validated methodologies	Q22	IV	5.0 ± 0.2	100.0%	0.0%	0.0%

**Table 22 jcm-14-04689-t022:** Result of the experts’ vote.

No.	Patient’s Recommendation	Rif Question	AHCPR	Mean ± SD	%		
					Agreement	Abstained	Disagreement
47	Patients’ suggestions must be considered in the drafting and periodic revision of the mesotherapy guideline	Q23	IV	4.4 ± 1.0	79.8%	16.0%	4.2%
48	Mesotherapy for analgesic purposes must be integrated into the individual path of care and assistance of the individual patient	Q23, Q6	IV	4.9 ± 0.3	100.0%	0.0%	0.0%
49	The patient has the right to be subjected to mesotherapy based on scientific evidence	Q23	IV	4.9 ± 0.4	99.2%	0.0%	0.8%

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
