# Peer review of "International Consensus Guidelines on the Safe and Evidence-Based Practice of Mesotherapy: A Multidisciplinary Statement"

_jcm, 2025, doi:10.3390/jcm14134689_

Round 1

Reviewer 1 Report

Comments and Suggestions for Authors

The submitted manuscript addresses a highly relevant and timely topic. The attempt to provide international consensus guidelines on mesotherapy is commendable and potentially valuable for clinicians and researchers. However, in its current form, the manuscript cannot be accepted without major revision. Below are detailed suggestions to improve the quality, clarity, and overall structure of the manuscript.

Language and Clarity: The manuscript requires significant improvement in language, grammar, and overall readability. The presentation of results and key statements is often unclear, making the manuscript difficult to follow. The authors should consider professional language editing to enhance clarity and ensure consistency of terminology throughout the text.

Structure and Presentation: The presentation of tables and figures needs to be simplified. Many of the table and figure titles are excessively detailed and repetitive with the main text, making them appear more as part of the narrative rather than as independent visual elements. The authors are recommended to shorten and standardize figure and table captions, focusing only on the most essential information.

Duplication of Labels: There is redundant labeling of several tables and figures. For example, in Table 8, the title is repeated both above and below the table:

Above: "Table 8. Studies on localized pain"
Below: "Table 8. Studies on localized pain. The table shows the number of mesotherapy sessions..."

This pattern is seen in other tables as well (e.g., Table 11) and should be corrected to avoid unnecessary repetition.

Image and Table Quality: The visual quality of several tables (especially Tables 11, 13, and 19) should be improved to ensure legibility and proper formatting for publication.

Permission for Reproduced Material: For Figures 4, 5, and 7, as well as Table 13, it is unclear whether appropriate permissions for reproduction have been obtained. The authors should confirm and provide documentation where necessary.

Statistical Reporting: It is recommended to include confidence intervals (CIs) for quantitative data wherever applicable, to improve the scientific rigor and interpretability of the findings.

While the manuscript has strong potential, the issues outlined above must be addressed comprehensively. A major revision is required before the manuscript can be reconsidered for publication

Comments on the Quality of English Language

The manuscript requires significant improvement in language, grammar, and overall readability. The presentation of results and key statements is often unclear, making the manuscript difficult to follow. The authors should consider professional language editing to enhance clarity and ensure consistency of terminology throughout the text

Author Response

We sincerely thank the reviewer for the valuable and constructive comments, which were very helpful in improving the quality and clarity of the manuscript. We have addressed each point as follows:

  • Language and Clarity
    The entire manuscript was professionally revised by the journal’s language editing service to improve grammar, terminology, and overall readability.
  • Structure and Presentation of Tables and Figures
    We reduced the number of figures and simplified figure details. Table formatting was revised following the graphical suggestions provided by the editorial service to improve clarity and consistency.
  • Duplication of Labels
    Redundant labeling above and below tables (e.g., Table 8 and Table 11) has been corrected to avoid repetition.
  • Image and Table Quality
    The visual quality of tables was enhanced. Low-resolution figures were replaced, and formatting was adjusted to meet publication standards.
  • Permission for Reproduced Material
    Figures 4, 5, and 7 were either canceled or replaced with original figures created by the authors to avoid any copyright issues. Table 13 was also revised accordingly.
  • Statistical Reporting
    Not applicable. No additional quantitative data were presented that required confidence intervals in this context.
  • Quality of English Language
    As mentioned above, the entire manuscript has been reviewed and revised by a professional English editing service to ensure consistency, accuracy, and clarity.

We are grateful for your detailed review, which has contributed significantly to improving the manuscript. We hope the revised version meets the standards for publication.

Reviewer 2 Report

Comments and Suggestions for Authors

Dear Authors,

Congratulations on your interesting and well-prepared manuscript. Your work addresses an important topic with clear clinical relevance. I appreciate the effort invested in this study.

I have a few suggestions that might help improve the manuscript further:

Comment 1: 
“Firstly, you could change the sentence in Introduction: ‘highlighted its benefits and limitations’ with ‘critically evaluated its clinical benefits and limitations.’ Also, you could change the sentence in Methods: ‘submitted for voting’ is somewhat unclear; better to say ‘submitted for structured evaluation and voting.’”

Comment 2:
“The nature of the results is described adequately: framework definition. It might be useful to provide an example or specify a field of application (e.g., pain management, sports medicine).”

Comment 3:
“In the Conclusion: Suggestion for improvement: Instead of the general phrase ‘can be addressed through scientific research,’ a more dynamic formulation would be: ‘These limitations underscore the need for continued high-quality research and scheduled guideline updates.’”

Comment 4:
“Abstract could be improved”

Author Response

We sincerely thank the reviewer for the thoughtful and constructive comments. We appreciate your recognition of the manuscript’s relevance and your helpful suggestions, which have been carefully addressed as follows:

  • Comment 1 – Wording in Introduction and Methods:
    We revised the sentence in the Introduction to read: "critically evaluated its clinical benefits and limitations".
    In the Methods section, we changed "submitted for voting" to "submitted for structured evaluation and voting".
    Revision completed.
  • Comment 2 – Clarification of fields of application:
    We have added specific examples (e.g., pain management, sports medicine) in both the abstract and the main text to clarify potential applications of the recommendations.
    Revision completed.
  • Comment 3 – Improved formulation in the Conclusion:
    We incorporated the proposed dynamic wording into the Conclusion section:
    "These limitations underscore the need for continued high-quality research and scheduled guideline updates."
    Revision completed.
  • Comment 4 – Abstract improvement:
    The abstract has been revised and improved accordingly. Final English editing is currently ongoing by a professional editor.Revision completed.
  • Language and Clarity
    The entire manuscript was professionally revised by the journal’s language editing service to improve grammar, terminology, and overall readability.
  • Structure and Presentation of Tables and Figures
    We reduced the number of figures and simplified figure details. Table formatting was revised following the graphical suggestions provided by the editorial service to improve clarity and consistency.

Thank you again for your valuable feedback, which has significantly improved the quality of our manuscript.

Reviewer 3 Report

Comments and Suggestions for Authors

i read with great interest the manuscript on mesotherapy guidelines as it is a very importnat topic- comments:

  • why international and not italian consesus?
  • -explain further line 187 (nor for the avoidance of nevi or other skin lesions)
  • correct the in-text citations style
  • explain better the "some" dermatological diseases that mesotherapy can be applied with well presented examples
  • generally well presented with clever and interesting questions

Author Response

We thank the reviewer for the thoughtful and constructive comments, which we found very helpful in improving our manuscript. Please find our responses below:

  • Why international and not Italian consensus?
    We have clarified this point in the manuscript: the consensus is international because all the representatives of the existing scientific mesotherapy societies worldwide participated in the development of the guidelines.
  • Line 187 – clarification regarding nevi and other skin lesions
    This sentence has been further clarified to ensure better understanding.
    Revision completed.
  • In-text citation style
    We have carefully reviewed and corrected the in-text citation formatting throughout the manuscript.
    Revision completed.
  • Clarification of dermatological conditions
    We appreciate this suggestion. Table 13 now includes well-documented examples of dermatological conditions where mesotherapy may be applied. We have also modified the title of Table 13 accordingly.
    Revision completed.

We are grateful for your positive overall evaluation of the manuscript and for recognizing the relevance of the questions addressed.

Round 2

Reviewer 1 Report

Comments and Suggestions for Authors

After revision manuscript can be considered for publications.